# Expression Analysis of Lipocalin 2 (LCN2) in Reproductive and Non-Reproductive Tissues of *Esr1*-Deficient Mice

**DOI:** 10.3390/ijms24119280

**Published:** 2023-05-25

**Authors:** Jan C. Kessel, Ralf Weiskirchen, Sarah K. Schröder

**Affiliations:** Institute of Molecular Pathobiochemistry, Experimental Gene Therapy and Clinical Chemistry (IFMPEGKC), RWTH University Hospital Aachen, D-52074 Aachen, Germany; jan.kessel@rwth-aachen.de

**Keywords:** estrogen receptor, Esr1/ERα, lipocalin 2 (LCN2), tissue-specific, reproductive tissue, non-reproductive tissue, ovary

## Abstract

Estrogen receptor alpha (ERα) is widely expressed in reproductive organs, but also in non-reproductive tissues of females and males. There is evidence that lipocalin 2 (LCN2), which has diverse immunological and metabolic functions, is regulated by ERα in adipose tissue. However, in many other tissues, the impact of ERα on LCN2 expression has not been studied yet. Therefore, we used an *Esr1*-deficient mouse strain and analyzed LCN2 expression in reproductive (ovary, testes) and non-reproductive tissues (kidney, spleen, liver, lung) of both sexes. Tissues collected from adult wild-type (WT) and *Esr1*-deficient animals were analyzed by immunohistochemistry, Western blot analysis, and RT-qPCR for *Lcn2* expression. In non-reproductive tissues, only minor genotype- or sex-specific differences in LCN2 expression were detected. In contrast, significant differences in LCN2 expression were observed in reproductive tissues. Particularly, there was a strong increase in LCN2 in *Esr1*-deficient ovaries when compared to WTs. In summary, we found an inverse correlation between the presence of ERα and the expression of LCN2 in testes and ovaries. Our results provide an important basis to better understand LCN2 regulation in the context of hormones and in health and disease.

## 1. Introduction

Despite increased consideration of sex as a biological variable in preclinical studies, sex bias in favor of males widely persists [1,2,3]. However, the situation has gradually improved in recent years due to specific requirements for research projects, e.g., to include equal numbers of male and female subjects [1]. This contributes to our knowledge of sex-specific interactions and improves the quality of research itself [1,2,3]. Sex-specific differences occur in all human organ systems and are majorly caused directly or indirectly by altered sex hormone levels [4,5]. In particular, due to their broad spectrum of actions, steroid hormones need to be studied beyond their reproductive functions.

Estrogens, especially 17β-estradiol (E2), form the main class of female sex hormones. They are primarily synthesized in the ovaries, and to a lesser extent in adipose, testis, and adrenal tissues [6,7]. Nevertheless, their functions are important in both males and females [6]. Estrogen signaling is predominantly mediated by the nuclear receptors termed estrogen receptor alpha (ERα) and estrogen receptor beta (ERβ), which are encoded by the *Esr1* and *Esr2* genes [6,8]. In addition, E2 mediates signaling via the membrane-bound G protein-coupled estrogen receptor 1 (GPER1) [8]. All these receptors exhibit specific expression patterns in a variety of tissues [6,8], which can differentiate between species and are topics of ongoing investigation [9].

ERα is expressed primarily in the liver, bone, mammary gland, uterus, ovary, and vagina [8]. ERα signaling is initiated in the cytoplasm by the binding of E2, followed by dimerization and receptor translocation into the nucleus [6]. The nuclear E2-bound ERα regulates various genes via estrogen-responsive elements (EREs) [6]. In particular, animal models have proven to be reliable tools for studying estrogen receptor interactions [10]. In 2002, Seth and colleagues found that the *Lcn2* promotor contains an ERE [11]. This and other studies indicate that LCN2 may be regulated by estrogens and consequently should show sex-specific expression differences [12].

*Lcn2* encodes for lipocalin 2 (LCN2), also known as neutrophil gelatinase-associated lipocalin (NGAL), and 24p3 lipocalin. LCN2 is a 25-kDa glycoprotein that has been linked to an increasingly broad range of immunological and metabolic functions [13,14,15]. Clinically, LCN2 belongs to the lipocalin family of proteins and is already considered an important marker for prognosis and diagnosis in acute kidney injury and various malignancies [12,16,17,18]. This multifaceted importance derives from its structure and associated functionality. Lipocalins are characterized by a β-barrel-like structure with a hydrophobic pocket that allows the binding and transportation of hydrophobic molecules, such as steroids, retinoids, or hormones [19]. Immunological functions of LCN2 are mainly based on its ability to sequestrate bacterial iron-bound siderophores, thereby lowering bacterial growth [20]. LCN2 is strongly secreted by neutrophil granulocytes during inflammatory conditions [19,21]. Furthermore, it is also secreted by other cell types, such as hepatocytes and adipocytes, e.g., as a messenger or carrier protein [12,22]. In summary, LCN2 is an important mediator in immunological and metabolic processes.

The function of LCN2 to transport the reproductive hormone precursor cholesterol suggests a link to the endocrine system [23,24]. This is highlighted by its expression in Leydig cells in the testes, which are the primary source of testosterone or androgens in males, and in epithelial cells of the uterus, which are highly sensitive to E2 [25,26]. Therefore, sex-specific changes need to be investigated to ultimately understand the regulation of LCN2. However, there are only a few studies that have addressed this topic so far. Some previous studies showed that LCN2 may even have profoundly different functions in the reproductive tissues of males and females.

In males, LCN2 possibly affects overall fertility, while in females, an immunological function is discussed [27,28]. In female mice, the natural estrous cycle has been shown to be a potential variable in LCN2–ERα interactions. Huang et al. described an estrous cycle-dependent expression of LCN2 in mouse uterine epithelial cells [28]. Nonetheless, reproductive tissues represent only a partial spectrum of the interactions between ERα and LCN2. Both murine and human non-reproductive organs such as liver and white adipose tissue have shown sex-specific differences linked to LCN2 expression [29,30]. We have recently reported that female *Lcn2*-deficient (*Lcn2^−/−^)* mice develop a more severe phenotype of fructose-induced steatosis compared to males [31]. Furthermore, others showed that E2 induces increased LCN2 expression in liver and adipose tissues in ovariectomized mice [32]. Regarding the current literature, mainly adipose and breast tissues have been analyzed for direct interactions with ERα and LCN2 [12,30,33]. Surprisingly, LCN2 correlates differently with ERα expression and depends on the type of tissue analyzed. This suggests that both sex- and tissue-specific factors play a role in its regulation. Consequently, no general rule can be established for the correlation of LCN2 and ERα expression.

To date, differences in LCN2 levels associated with ERα expression have not been systematically investigated yet. Therefore, the present study focused on different reproductive (testis, ovary) and non-reproductive (lung, liver, kidney, spleen) organs in which LCN2, ERα, or both proteins are known to have profound effects. We examined the expression of these proteins in the tissues of male and female wild-type (WT) animals and compared them with the expression in tissues of a conventional *Esr1* knockout mouse strain generated in 2003 [34]. The obtained findings indicate a differential tissue-specific expression of LCN2 in *Esr1*-deficient animals because LCN2 levels strongly increase in reproductive tissues when ERα is absent.

## 2. Results

### 2.1. Characteristics of the Esr1 Knockout Mouse Phenotype

Estrogen receptor 1 knockout (*Esr1*^−/−^) mice are widely used in research to study aspects of estrogen receptors and their functions [8,10]. Sex-specific differences and estrogen receptor signaling are the subject of ongoing investigation [7,35]. To validate global deletion of *Esr1* in the used mouse model (B6N(Cg)-*Esr1^tm4.2Ksk^*/J), DNA was isolated from ear punches and subjected to genotyping. In our genotyping protocol, the WT allele produced an amplicon of 287 bp in size with the chosen primer combination, while DNA taken from *Esr1*-depleted mice resulted in the amplification of a 550 bp fragment, which is due to the insertion of a floxed TK-neo selection cassette (Figure 1A). The presence of both fragments was observed in heterozygote animals, carrying a WT and a knockout allele.

*Esr1*^−/−^ phenotype characteristics have been described in various studies [8,34]. Most strikingly, female *Esr1*-deficient mice exhibit impaired development of reproductive organs. In particular, uteri of respective mice are hypoplastic compared to WTs, while knockout ovaries further develop distinct haemorrhagic, cystic follicles (Figure 1B,D) [8]. In addition, total body weight, which is a general readout for the murine metabolism [36], showed a 1.2-fold increase in *Esr1*-depleted females when compared to female WT littermates, WT males, or *Esr1*-deficient males (Figure 1C).

As mentioned above, LCN2 is a secretory protein which is secreted by various cell types, such as hepatocytes and neutrophil granulocytes [12,22]. We found significantly increased levels of LCN2 in the serum of male *Esr1*-deficient animals when compared to WT males (Figure 2A). In contrast, the LCN2 serum quantities were comparable in female WT and *Esr1*-depleted females (Figure 2B), suggesting that the content of LCN2 follows a sex-specific trend in mice.

### 2.2. ERα and LCN2 Expression in Liver Tissue

ERα is known to mediate protective functions in the liver. This is underlined by increased hepatic fat accumulation in female *Esr1*-deficient mice compared to WT mice when challenged with a high-fat diet [37]. In our study, adult female *Esr1*-deficient mice exhibited a 1.3-fold average increase in liver weight compared to female WT animals, while WT and *Esr1*-deficient males developed similar weights compared to *Esr1*-deficient females (Figure 3A).

Although sex-specific differences in hepatic ERα have previously been reported, a direct link to LCN2 has not been described yet. For this reason, we examined the hepatic expression of LCN2 in *Esr1*-deficient and WT animals by immunohistochemistry. We detected single LCN2-positive cells (punctual) and areal expression sites (Figure 3B). However, no clear differences between sexes or genotypes could be observed in the staining. This is in line with Western blot analysis showing no significant difference between male *Esr1*-deficient and wild-type animals (Figure 3C). In addition, female livers showed strong variations in LCN2 expression, irrespective of whether ERα was present or not (Appendix A).

Next, reverse transcription and quantitative real-time PCR (RT-qPCR) analysis was used to compare the hepatic mRNA expression of *Lcn2* in WT and *Esr1*^−/−^ females and males (Appendix A). Although we observed a strong variation in female WT mice, respective animals exhibited significantly increased LCN2 expression compared to female *Esr1*-deficient mice and males that both showed similar expression of LCN2.

In contrast, sex-specific differences were observed in hepatic ERα expression (Figure 3D). In liver tissues, female WTs showed stronger staining intensities compared to male animals. *Esr1*-depleted tissue sections and IgG negative controls remained negative for ERα with low background staining. In addition, Western blot analysis of ERα confirmed staining results (Appendix A).

In summary, our data show differences between the RNA and protein analyses and do not provide a clear conclusion regarding whether the presence of ERα directly or indirectly affects the level of LCN2 in male or female livers.

### 2.3. LCN2 Expression in Esr1-Deficient Lung and Kidney Tissue

In the lung, LCN2 expression is primarily limited to type II pneumocytes, fibroblasts, macrophages, and neutrophil granulocytes [38,39]. ERα is not expressed in healthy lung tissue but is frequently detected in lung carcinoma cells [40]. Immunohistochemical detection for LCN2 was performed to localize LCN2 tissue distribution. We observed single LCN2-positive (punctual) and area-specific staining signals (Figure 4A). However, staining results showed no clear sex- or genotype-specific difference in LCN2 expression in the lungs. Therefore, Western blot analyses were performed, which demonstrate significantly lower expression of pulmonary LCN2 in *Esr1*-deficient females compared to female WTs. On the other hand, male animals showed LCN2 expression that was similar to female WTs (Appendix A). Furthermore, our results confirmed that ERα was not detectable in the lung tissue of either females or males (Figure 4B and Appendix A).

LCN2 is a well-established prognostic and diagnostic marker in acute kidney injury [16,41]. It is primarily expressed in epithelia cells of renal nephron tubule segments, Henle’s tubules, and collecting ducts [41]. As previously described, ERα is primarily expressed in sex organs and not expressed in healthy kidney tissue [42]. In our immunohistochemistry, we observed weak LCN2 staining of single cells (punctual) and some faint areal signals in all kidney samples in immunohistochemistry without any clear sex- or *Esr1*-specific differences (Figure 4C). However, Western blot analysis showed decreased LCN2 levels in female *Esr1*-depleted kidneys compared to WT animals (Figure 4D), which was not statistically significant (*p* = 0.059). ERα protein expression could not be detected in kidney tissue, while LCN2 expression in male kidneys showed strong variations in both genotypes (Appendix A).

The spleen, as the biggest secondary lymphoid organ, is entrusted with important immunological and phagocytotic functions and enriched with neutrophil granulocytes [43,44]. However, analyzed spleen tissue showed no sex- or genotype-specific differences in LCN2 expression (Appendix A) or changes in ERα between female and male WTs (Appendix A).

In conclusion, non-reproductive organs such as the lung and kidney, which physiologically do not express ERα, showed genotype differences in LCN2 expression. These findings suggest that underlying systemic parameters in *Esr1*-deficient mice cause these changes.

### 2.4. Esr1^−/−^ Testes Show Increased LCN2 and Steroidogenic Acute Regulatory Protein Expression

ERα is involved in key functions of the male urogenital system [45]. Both male and female homozygous *Esr1*-deficient mice are infertile [34]. However, the reproductive tracts of female animals already show macroscopic changes that indicate a reproductive defect (cf. Figure 1B), whereas the male reproductive tract appears normal (cf. Appendix A [34]. Histological examination of *Esr1*-depleted males showed that they develop distinct histological morphologies in testes, efferent ductuli, and epididymis [45]. In contrast, the effects of LCN2 in the male reproductive tract remain poorly understood. Studies show that testicular LCN2 expression increases with age and is primarily limited to Leydig cells [25], which express ERα as well. In addition, Leydig cells hold important functions in the production of sex hormones and the regulation of spermatogenesis [45]. To determine possible interactions between LCN2 and ERα in testis, we next compared *Esr1*-deficient tissues with WT samples.

First, total testis size, total testis weights, and calculated testis-to-body weight ratios showed no significant differences between the two genotypes (Appendix A). However, more detailed histological analysis showed that the seminiferous tubule lumens of *Esr1*^−/−^ animals were dilated and appeared almost “empty” (Figure 5A). *Esr1*^−/−^ interstitial segments seemed hyperplastic and magnified compared to WT segments. Immunohistochemical detection of LCN2 showed positive signals in interstitial Leydig cells and highly increased LCN2 expression in *Esr1*-deficient testes (Figure 5B, upper panels). Furthermore, ERα immunohistochemistry showed strong ERα expression in the interstitial Leydig cells of WT testes, whereas the IgG negative controls and *Esr1*^−/−^ testes remained negative (Figure 5B, lower panels).

These staining results were supported by immunofluorescence double staining for LCN2 and steroidogenic acute regulatory protein (STAR). STAR, which facilitates the rapid movement of cholesterol from the outer to the inner mitochondrial membrane, is highly expressed in Leydig cells [19,46,47]. LCN2 (Alexa Fluor 488-labeled, *green*) and STAR (Alexa Fluor 555-labeled, *red*) showed positive staining signals in interstitial segments in both WT and *Esr1*-deficient males (Appendix A). Strikingly, *Esr1*-deficient testes displayed an increased LCN2 immunofluorescent signal compared to WTs. This finding was confirmed by Western blot analysis showing significantly decreased LCN2 expression in WT testes compared to *Esr1*-deficient males (Figure 5C). In addition, all analyzed *Esr1* knockout testes showed increased STAR protein expression compared to WTs in Western blot analysis. RT-qPCR analysis confirmed significantly increased *Star* and *Lcn2* mRNA expression in *Esr1*-deficient testes compared to WTs (Figure 5D), demonstrating an inverse correlation between the expression of LCN2 and ERα in testicular tissues from male mice for the first time. This highlights the importance of studying the expression and regulation of LCN2 in reproductive tissues.

### 2.5. Increased LCN2 Expression in Esr1-Deficient Ovaries

Estrogen, the major female reproductive hormone, is associated with a broad range of functions, such as affecting gene expression in reproductive organs [48]. Serum estrogen levels in female mice vary naturally with the estrous cycle [6,49], which is regulated by the hypothalamic–pituitary–gonadal axis [6]. Studies in mice showed that different proteins in uterine tissue are dynamically regulated depending on the estrous phase, including LCN2 [28,50].

In our study, female mice were histologically assigned to the most likely stage of the estrous cycle by analyzing H & E staining of vagina, uterus, and ovary tissues (Appendix A). For this purpose, the specific different histological features of all three organs were used (Appendix A). Immunohistochemical detection of LCN2 in WT uteri showed differential expression depending on the stage of the estrous cycle. Females in proestrus and estrus phases presented stronger LCN2 expression compared to animals in the metestrus or diestrus stage (Appendix A). Very low to virtually no LCN2 expression was observed in *Esr1*-deficient uteri, underlining a connection between ERα and LCN2 in uterine tissue.

In contrast to LCN2 expression in the uterus, very little is known about its physiological role in ovarian tissue [51,52]. Clinically, LCN2 is discussed in the context of polycystic ovary syndrome (PCOS) and ovarian cancer [53,54,55,56]. ERα is strongly expressed in the murine ovary but limited to interstitial theca and stromal cells [8]. To date, nothing is known about interactions between ERα and LCN2 in the ovary. Therefore, we next investigated whether there is a link in the murine ovary. To do so, reproductive tissue sections were first subjected to routine H & E staining. *Esr1*-deficient ovaries exhibited haemorrhagic cystic follicles, as previously described (Figure 6A) [34]. Immunohistochemical staining confirmed interstitial and stromal cells as sites of nuclear ERα expression (Figure 6B, right panels). In WT ovaries, isotope-specific IgG negative controls showed low background signals, whereas unspecific staining (red and brown) occurred when *Esr1*^−/−^ ovaries were stained with the IgG negative control. Immunohistochemical localization of LCN2 in WT ovaries showed scattered, punctual, and positive staining, mostly located in stromal tissue (Figure 6B, left panels). In contrast, *Esr1* knockout ovaries displayed markedly increased quantities of LCN2-positive cells, which were mainly restricted to stromal interstitial tissue. Additionally, ovarian tissue samples were analyzed for LCN2 and STAR expression by immunofluorescence double staining. As in Leydig cells of testis tissue, STAR is a key factor in theca and granulosa cells related to steroid hormone production in the ovary [57]. Our staining confirmed expression in interstitial cells of the ovary (Figure 6C). Interestingly, ovaries taken from female WTs showed very low LCN2 staining, which was distributed to single small cells, while *Esr1* knockout ovaries showed strong LCN2 expression in large cell clusters (Figure 6B,C). Additionally, WT ovaries showed a slightly stronger STAR signal compared to *Esr1*^−/−^ animals.

Analysis of mRNA expression by RT-qPCR was performed with ovarian tissue samples from all female animals to substantiate previous results. *Esr1* knockout ovaries showed an almost 300-fold increase in *Lcn2* mRNA expression compared to WT animals (Figure 6D). However, *Star* mRNA expression showed only a minor trend of increased expression in WTs compared to *Esr1* knockout ovaries. The massive increase in *Lcn2* expression might indicate an inflammatory reaction in *Esr1*-depleted ovaries. Consequently, we analyzed the expression of tumour necrosis factor-α (*Tnf*), integrin αM (*Itgam*), myeloperoxidase (*Mpo*), and interleukin 1β (*Il1b*) through RT-qPCR. *Tnf* and *Itgam* expression levels were significantly increased in *Esr1* knockouts compared to WT ovaries, while the mRNA expression levels of *Mpo* and *Il1b* tended to be elevated in knockouts compared to WTs (Appendix A).

In summary, we found an inverse relation between LCN2 and ERα expression in murine ovarian tissue. *Esr1*-deficient ovaries showed a great increase in LCN2 expression compared to very low levels in WT animals. This was accompanied by a significant increase in the expression of inflammatory markers such as *Tnf* and *Itgam.*

## 3. Discussion

LCN2 is involved in various metabolic and immunologic processes [29]. Accumulating evidence suggests that these functions may be differentially regulated by ERα activity depending on sex and tissue type [12]. This evidence originates from individual studies using multiple animal models and the application of different methods to fit the respective scope of research. Adipose and breast tissues are the most extensively studied organs to date when investigating direct interactions between LCN2 and Erα [12]. These studies have shown that LCN2 is regulated in a tissue-specific manner depending on estrogen and its receptors. However, due to limited research on other organs and tissues, these findings are difficult to place in a broader context. In addition, sex- and tissue-specific regulation of LCN2 is only partially understood. To further elucidate Erα and the potential sex-dependent expression patterns of LCN2, we examined its expression in non-reproductive and reproductive organs of a complete *Esr1*-deficient model and compared it with WT controls.

Total body weights of *Esr1* knockout mice show a loss of sexual dimorphism [34,58]. Consistent with these studies, we confirmed adult female *Esr1* null mice (Figure 1C) were significantly heavier than aged-matched wild-types, while the body weights of males were not affected by the absence of ERα. Hewitt and co-workers found a decrease in bone density with a concomitant increase in body fat percentage in *Esr1*-deficient males [34], which may explain the unchanged total body weight of males. Importantly, female Esr1 null mice had fulminant alterations to their reproductive tracts (Figure 1B,D).

LCN2 is a secretory protein that is expressed by various cell types and is detectable in the serum samples of mice and humans [12,30]. Western blot analysis showed higher LCN2 serum levels in *Esr1*^−/−^ males compared to WTs and female mice (Figure 2). Because hepatocytes and adipocytes are the primary secretory sites of LCN2, the altered function of these cell types could explain the observed trend [12]. In addition, others already identified sex and weight as variables influencing LCN2 secretion or its circulating levels [59,60]. In line with these findings, previous studies have proposed a link between serum LCN2 levels and its expression in liver tissue [12,30,32,59].

Immunohistochemistry detection of LCN2 showed areal- and punctual-positive staining in the liver tissue (Figure 3B), most likely resulting from LCN2-positive hepatocytes and leukocytes [61,62]. Based on other studies and our results, we initially speculated about whether there is a possible link between the level of LCN2 in serum and hepatic expression. However, our results do not confirm this because, in contrast to serum expression patterns, *Esr1* knockout male livers showed no significant difference in LCN2 expression compared to WTs (Figure 3C), which was in line with no change in mRNA levels (Appendix A). In general, we observed strong variations in hepatic LCN2 expression in female WT and *Esr1*^−/−^ mice (Appendix A). Regarding mRNA level, female *Esr1*-deficient females showed significantly lower expression of *Lcn2.* However, this change was not seen in the protein levels of LCN2, which might be due to heterogeneity in mRNA translation [63]. In addition, Della Torre and colleagues showed that the estrous cycle in female mice influences the expression of hepatic genes [64]. This would also be conceivable for *Esr1* and *Lcn2* in female livers, as it has also been detected in uterine tissue [28]. However, to prove this hypothesis, further studies with more animals per estrous phase are required.

Western blot analysis and immunohistochemistry detection of hepatic ERα validated previous findings of distinctly increased expression levels in female WTs compared to males despite the discussed variation (Figure 3D and Appendix A) [65]. Lastly, the analysis of total liver weights showed a similar distribution compared to the data regarding total body weights (Figure 1C and Figure 3A). Again, this is consistent with previous studies and could be explained by gradual fat accumulation in female *Esr1*-deficient livers [12,34]. A possible explanation for the lack of accumulation in male livers might be the sex dimorphism of protective functions of ERα. Although these are described in both sexes (also specifically in the liver), they could be compensated to a different extent upon deletion of ERα [66]. However, no clear difference in fat content was found in histological evaluation of the immunohistochemistry trials. It is possible that this difference is more pronounced in older animals [67] and should therefore be investigated in further studies.

Under normal physiological conditions, lung tissue does not express ERα [40]. Nevertheless, LCN2 and estrogen have significant pulmonary effects [38,40]. Previous studies show that LCN2 is an important immunological modulator in organs with contact surfaces to the external environment, such as the lung [26,38]. For this purpose, it is expressed by various cell types [38]. Furthermore, studies in *Lcn2*-deficient mice have detected impaired pulmonary immune functioning [38]. Immunohistochemical staining is suitable to investigate the protein expression and distribution of LCN2 in the lung to draw initial conclusions about the cell types expressing it. Single positive signals most likely originate from LCN2-expressing leukocytes, whereas areal-positive staining is possibly indicative of pneumocytes (Figure 4A) [38]. Western blot analysis showed significantly lower LCN2 expression in female *Esr1*-deficient lungs compared to similar levels detected in WTs and male tissues (Figure 4B). Due to the lack of ERα expression in lung tissue, altered systemic parameters such as E2 levels might possibly explain this observation [40]. Moreover, E2 can be bound by multiple receptors besides ERα, e.g., by ERβ, which has been detected in the lung tissue of mice [40]. Ultimately, elevated serum E2 levels in female *Esr1*-deficient mice may inhibit LCN2 expression through ERβ signaling in the lung. A similar inhibitory cascade was previously described in human adipose tissue by Kamble and colleagues [30]. Lower E2 levels in male knockouts compared to WT animals could explain why male *Esr1*-deficient mice tend to exhibit lower LCN2 expression compared to WTs (Appendix A).

Next, we focused on kidney tissue, another non-reproductive organ that does not physiologically express ERα [42]. Our aim was to analyze whether organs sharing this characteristic showed similar expression patterns of LCN2. Immunohistochemistry detection presented positive signals in all kidneys (Figure 4C). Leukocytes and tubular cells have been described as primary sources of renal LCN2 expression [16]. Areal-positive staining likely originated from tubular cells, whereas single positive signals possibly derived from migrating leukocytes. We observed that kidneys taken from *Esr1*-deficient females tended to exhibit lower LCN2 expression compared to WT and male animals (Figure 4D and Appendix A). Interestingly, previous studies have detected ERβ expression in the kidney cells of mice and humans [42,68]. Comparable characteristics regarding LCN2 and ERβ in lungs and kidneys possibly indicate similar regulation in both organs. The inflammatory and regenerative functions of LCN2 in both organs underline this theory [41]. Ultimately, our results suggest that non-reproductive organs which physiologically do not express ERα possibly share a similar LCN2 expression pattern. It is possible that this is caused by E2 via ERβ signaling.

In contrast to non-reproductive tissues, reproductive organs are less studied in terms of potential interactions with LCN2 and ERα. Nevertheless, there is growing evidence that LCN2 plays an important role in the female and male reproductive tracts [26,27]. In the present study, histological analysis of testis specimens showed hypertrophic interstitial segments in *Esr1*-deficient mice (Figure 5A). This is likely due to enhanced metabolic activity, more specifically raised serum testosterone and luteinizing hormone (LH) levels [25,34,69] in *Esr1* knockout compared to WT animals. Testosterone is primarily synthesized in Leydig cells, which are located in interstitial testicular tissue [70]. Increased LH in *Esr1*-deficient mice might additionally trigger hormone synthesis [34]. Supporting findings have been previously described in a similar mouse model by Rosenfeld et al. [69].

Leydig cells are known from the literature as LCN2-expressing cells and also as ERα-positive cells [25,70]. Therefore, we initially speculated that depletion of Esr1 could affect LCN2 expression in these cells. Interestingly, immunohistochemistry detection of LCN2 and ERα demonstrated drastically stronger LCN2 expression in *Esr1* knockout interstitial cells (Figure 5B). Western blot (Figure 5C) and RT-qPCR analysis (Figure 5D) proved a significant increase in LCN2 expression in *Esr1* null testes, while total testes weight was not altered (Appendix A). This finding suggests a link between LCN2, estrogen, and fertility in males. Consistent with this, busulfan-treated mice that had apoptosis of germ cell populations showed increased LCN2 expression in testis [25].

ERα deficiency enhances androgen biosynthesis in mouse Leydig cells [70]. Immunofluorescent staining in testis tissue confirmed the co-expression of STAR and LCN2 in interstitial segments (Appendix A). By using different methods, we demonstrate an increase in LCN2 expression in *Esr1*-deficient testis at both the mRNA and protein levels. This suggests a strict inverse correlation between ERα and LCN2 in testicular tissue and confirms ERα as a modulator of steroid synthesis in males.

LCN2 is known to be essential for fertilization through modulation of the membrane properties of the sperm in the uterotubal junction [27], but little is known about the precise function of LCN2 in the female reproductive tract. Previous rodent studies focused on the expression of LCN2 in uterine tissue, especially under the influence of hormonal changes during estrous phases, implantation, early pregnancy, and around birth [26,28]. Our data confirmed an estrous cycle-dependent LCN2 expression pattern in uterine epithelial cells, with higher levels in proestrus and estrus phases than in the metestrus or diestrus phase, as previously reported by Huang and colleagues [28]. Due to the fact that the *Esr1*-deficient animals lack natural ovarian cyclicity and exhibit the characteristics of a permanent diestrus stage [71,72], we speculated that LCN2 expression is also altered in these females. Indeed, we could show for the first time that uterine LCN2 expression is strongly decreased in all analyzed *Esr1* knockout compared to WT females (Appendix A). This confirms a positive correlation with LCN2 depending on the presence of ERα/*Esr1* in the uterus [26]. The mere fact that LCN2 can be regulated in a cyclic and organ-specific manner in female animals suggests fundamental differences between both sexes. Consequently, the strong fluctuations we found in the different non-reproductive organs in female animals could be related to hormonal changes during the different estrous stages. Further comprehensive studies with a more extensive number of female animals at each estrous stage are needed to determine whether LCN2 expression is indeed affected by the estrous cycle in other tissues.

To generate a more comprehensive overview of the female reproductive system, we next analyzed interactions between ERα and LCN2 in the ovary (Figure 6). In general, the expression and function of LCN2 in the ovary is poorly understood. To date, the role of LCN2 in PCOS in ovarian tissue is the best-studied link that might provide information about the ovarian functions of LCN2 [53]. PCOS is one of the most common causes of infertility in females and is characterized by metabolic alterations, which partially resemble the phenotype and characteristics of *Esr1*-deficient mice in regard to elevated serum E2 and LH levels [73,74,75].

In the present study, the phenotype of the *Esr1* knockout ovaries was analyzed in H & E staining (Figure 6A), which confirmed the findings of a previous report [8]. Using immunohistochemistry, we have shown for the first time that LCN2 expression in interstitial ovarian tissue is dramatically increased in *Esr1*-deficient compared with WT animals (Figure 6B). In line with this finding, RT-qPCR analysis showed about 300-fold higher quantities of *Lcn2* mRNA in *Esr1* null mice than in WTs (Figure 6D). Since only single LCN2-positive cells were detectable in WT ovaries, whereas larger cell areas were stained in *Esr1*-deficient ovaries, it could be assumed that the LCN2-expessing cell types in the knockout ovary are different than those in WTs.

Since STAR is a key factor for steroid hormone production in the ovary [57], it was expected to be expressed in both genotypes. However, we observed co-expression of STAR and LCN2 only in the ovaries of *Esr1*-deficient animals. In addition, we found strong variation in *Star* expression within the WT ovaries, and *Esr1* knockouts tend to express lower overall levels. Studies conducted in humans reported that *Star* expression is strongly upregulated during the luteal phase of the menstrual cycle [76]. Furthermore, studies in female mice showed trends of varying mRNA expression levels depending on the estrous cycle phase [77]. This could possibly explain the observed changes in STAR expression between *Esr1*-deficient and WT mice. Nevertheless, the molecular mechanisms underlying the inverse dependence between the presence of ERα and LCN2 are still unknown.

Interestingly, Burns and colleagues demonstrated a similar LCN2 expression pattern in a follicle-stimulating hormone (FSH) β (*Fshb*) knockout mouse model [78]. In this model, knockout mice also showed dramatically increased LCN2 expression originating from interstitial cells [78]. In contrast to *Esr1*-deficient mice, *Fshb* knockouts exhibited normal E2 serum levels; conversely, *Esr1* knockouts were associated with normal FSH levels [75,79]. Overall, this suggests that LCN2 is regulated by FSH and ERα in ovaries.

In addition, both metabolic and inflammatory functions in other tissues have been detected in connection with LCN2 and provide starting points for further analyses in the ovary [22,33]. RT-qPCR results in the present study showed that some inflammatory markers were significantly elevated in *Esr1*-deficient ovaries compared to WT controls (Appendix A). This could be indicative of an inflammatory function, as described in other tissues [22].

Overall, our study presents an overview of ERα and LCN2 expression patterns in the various non-reproductive tissues of a global *Esr1*-deficient mouse model, including male and female animals. In addition, the newly detected inverse expression pattern between ERα and LCN2 provides an entirely new approach to the expression of adipokines in female reproductive organs and potentially associated diseases. Our study provides evidence that LCN2 has important biological functions in female as well as male reproductive organs and should be studied in detail in the context of health and disease. However, it should be critically noted that we used complete *Esr1* knockout mice for our study, meaning that we cannot be sure if observed expression differences are of primary or secondary origin. To better address primary organ-specific effects, it might be more accurate to analyze mice conditionally disrupted for *Esr1* in the organ of interest.

## 4. Materials and Methods

### 4.1. Animal Husbandry and Tissue Collection

*Esr1*-deficient (B6N(Cg)-*Esr1^tm4.2Ksk^*/J) mice were generated as described by Korach and colleagues [34] and obtained from The Jackson Laboratory (JAX stock #026176, The Jackson Laboratory, Bar Harbor, ME, USA). All experiments were approved by the internal Institutional Review Board of the RWTH University Hospital Aachen (permit no.: TV40138). The animals were handled in compliance with German animal welfare law (Tierschutzgesetz, TSchG) and according to Directive 2010/63/EU. Mice were kept under 12 h light–dark conditions and had free access to normal food and water ad libitum. All animals used in this study were of reproductive age (mature adult) and between 12 and 15 weeks old when euthanized by cervical dislocation with preceding isoflurane sedation. Total body, whole liver, and testicular weights were documented for each animal. Reproductive and non-reproductive organs (testis, uterus, ovary, liver, lung, kidney, spleen) were dissected and prepared for further analysis (histology, protein, and RNA). Blood samples were collected using a Serum Microvette^®^ (#20.1291, Sarstedt, Nümbrecht, Germany) and stored at −80 °C. In this study, n = 8 male and n = 6 female *Esr1*-deficient mice were included, as well as n = 8 male and n = 9 female WT mice.

### 4.2. Genotyping

Genotyping was performed with DNA samples prepared from ear punches stored at −20 °C after biopsy collection. In brief, tissues were digested in lysis buffer containing 100 mM Tris-HCl, 200 mM NaCl, 5 mM ethylenediaminetetraacetic acid (EDTA), and 0.2% (*w*/*v*) sodium dodecyl sulphate (SDS) in H_2_O supplemented with proteinase K (0.1 mg/mL) and incubated at 55 °C overnight under shaking. The next day, samples were centrifuged for 10 min at 15,000× *g* at room temperature (RT) and the supernatant was supplemented with isopropanol (1:1). After incubation for 10 min at RT, samples were centrifuged (15,000× *g*). Pellets were washed with 70% ethanol, centrifuged (10 min, 15,000× *g*, RT), and finally resuspended in H_2_O. For *Esr1* genotyping, the DNA was used as a template in a standard genotyping PCR according to the protocol recommended by The Jackson Laboratory [80]. Briefly, the purified DNA was given to a master mix solution that contained reagents of the Taq PCR Core Kit (#201225, Qiagen, Hilden, Germany), RNase free H_2_O, primers (primer 1 (22055): CTG CCA AGG AGA CTC GCT AC (WT forward); primer 2 (22056): CCA CTT CTC CTG GGA GTC TG (common); primer 3 (22057): ATC CCA TGT GCT TGA GTG GT (MUT forward)), and dNTP mix. The amplification was conducted as follows: 30 cycles at 95 °C for 5 min, 95 °C for 1 min, 55 °C for 1 min, 72 °C for 3 min, and 72 °C for 10 min. Amplification was achieved on a T3000 Thermocycler (Biometra GmbH, Jena, Germany), and fragments were separated in 1.6% agarose gels supplemented with ethidium bromide for 40 min (90 V) using 1× TAE (40 mM Tris base, 20 mM acetic acid, and 1 mM EDTA) as a running buffer. Results were visualized in a GEL iX20 imager (Intas Science Imaging Instruments GmbH, Göttingen, Germany). Amplicon sizes were 287 bp (WT allele) and ~550 bp (mutant allele), respectively.

### 4.3. Protein Analysis

Isolated tissue was resuspended on ice in RIPA buffer containing 20 mM Tris-HCl (pH 7.2), 150 nM NaCl, 2% (*w*/*v*) NP-40, 0.1% (*w*/*v*) SDS, and 0.5% (*w*/*v*) sodium deoxycholate supplemented with the Complete^TM^ mixture of phosphatase inhibitors (#P5726-1ML, Sigma-Aldrich, Taufkirchen, Germany) and stored at −80 °C. Tissue samples were homogenized in an MM400 mixer mill (Retsch GmbH, Haan, Germany) as previously described [81]. The DC protein assay (Bio-Rad Laboratories GmbH, Düsseldorf, Germany) was used to quantify total amounts of protein in each sample. Samples were then supplemented with NuPAGE lithiumdodecyl sulfate (LDS) sample buffer and dithiothreitol (DTT). Equal protein amounts were heated at 80 °C for 10 min and loaded on 4–12% Bis-Tris gradient gels (#WG1401BX10, Thermo Fisher Scientific Inc., Waltham, MA, USA) with 2-(*N*-morpholino) ethanesulfonic acid (MES) as a running buffer. For analyzing serum samples, 1 µL of serum was loaded into each lane. Proteins were transferred to Protran^®^ Western blotting membranes (#10600002, 0.45 µm, GE Healthcare, Buckinghamshire, UK) in NuPAGE buffer containing 10% methanol. Equal loading and successful protein transfer was visualized by Ponceau S staining. Subsequently, unspecific binding sites were blocked by 5% (*w*/*v*) non-fat milk powder in Tris-buffered saline supplemented with 0.1% Tween 20 (TBS-T). Membranes were incubated with primary antibodies (Appendix A) that were diluted in 2.5% (*w*/*v*) non-fat milk powder in TBS-T. Incubation was conducted on a shaker at 4 °C overnight. For visualization of protein signals, horseradish–peroxidase-conjugated secondary antibodies were used in combination with the SuperSignal chemiluminescent substrate (#34076, Thermo Fisher Scientific). Membranes were re-probed with either glyceraldehyde-3-phophate dehydrogenase (GAPDH) (in lungs, livers, and testes) or vinculin (in kidneys) [82] to demonstrate equal protein loading for each gel. Transferrin was used as a protein loading control in serum samples as previously described by Minagawa and colleagues [83].

### 4.4. RNA Analysis

Dissected tissue samples were snap-frozen in liquid nitrogen and stored at −80 °C until further processing. Tissue was re-suspended in RNA lysis buffer with DTT and homogenized in an MM400mixer mill (Retsch GmbH, Haan, Germany) as described previously [81]. RNA extraction was performed using the PureLink RNA Mini kit (#12183018A, Thermo Fisher Scientific Inc.) including DNAse digestion (#12185-010, Thermo Fisher Scientific) according to the manufacturer’s guidelines. The first strand of complementary DNA (cDNA) was synthesized from 1 µg of RNA (800 ng for ovaries) with Superscript II reverse transcriptase (#18064-014, Thermo Fisher Scientific) and random hexamer primers (#C118A, Thermo Fisher Scientific). cDNA samples were diluted in RNase-free H_2_O and stored at −20 °C as previously described [17]. For reverse transcription and quantitative real-time PCR (RT-qPCR), 5 µl of cDNA and specific primers (Appendix A) were given in a total volume of 25 µL SYBR-Green^TM^ qPCR SuperMix (#56465, Thermo Fisher Scientific). Consecutive cycle conditions were initial denaturation at 95 °C for 10 min, followed by amplification for 40 cycles at 90 °C for 15 s and 60 °C for 1 min. All samples were loaded in duplicate. mRNA expression was normalized to either the expression of β-actin (*Actb,* in testes and ovaries) or ribosomal protein S6 (*Rps6*, in livers). Final mRNA quantities were calculated with the 2^−ΔΔCT^ method [84].

### 4.5. Histology

#### 4.5.1. Hematoxylin and Eosin (H & E) Staining

Fresh tissue samples were fixated for 24 h at 4 °C in 4% neutral buffered formaldehyde (stabilized with methanol), dehydrated, and embedded into paraffin. Subsequently, 3 µm thick sections were prepared, deparaffinized in xylene, and subjected to decreasingly graded ethanol for rehydration. Samples were stained with hematoxylin (#CS70030-2, DAKO, Agilent Technologies, Santa Clara, CA, USA) for 12 min and were afterwards placed under running tap water for a further 10 min. Slides were incubated in Eosin at a pH of 4.5 (#HT110216, Sigma-Aldrich) for 30 s and dehydrated in increasingly graded ethanol and xylene. DPX was used as a mounting medium (#06522, Sigma-Aldrich). The H & E stain was used for estrous cycle staging as previously described [49,80,85]. Therefore, specific histological features of female reproductive tissues were used to distinguish between the different estrous stages, as summarized in Appendix A.

#### 4.5.2. Immunohistochemistry Staining

Formalin-fixed and paraffin-embedded tissues were deparaffinized and prepared as described above. Heat-induced antigen retrieval was performed by placing the slides in citrate buffer (10 mM, pH 6.0, 0.05% Tween 20) for 30 min in a steamer, followed by cooling on ice for 30 min. Subsequently, samples were washed in phosphate buffered saline (PBS) and PBS supplemented with 1% Tween 20 (PBS-T). For immunohistochemical detection of ERα and LCN2, endogenous biotin and avidin binding sites were blocked with a biotin blocking system (#X0590, DAKO, Agilent Technology, Santa Clara, CA, USA). Furthermore, unspecific antibody binding sites were blocked at RT for 90 min in 5% normal serum from the source species of the secondary antibody (rabbit serum, #X0902, DAKO) in PBS supplemented with 1% bovine serum albumin, 0.1% cold fish skin gelatine, 0.1% Triton X-100, and 0.05% Tween 20. Primary antibodies were diluted in blocking solution (LCN2: #AF3508, R&D Systems, Inc., Minneapolis, MN, USA, 1:40; ERα (clone 6F11): #MA1-27107, Thermo Fisher Scientific, 1:50) and incubated on samples at 4 °C overnight. IgG negative controls (LCN2: normal goat IgG control, #AB-108-C, R&D Systems; ERα: negative control mouse IgG_1_, #X0931, DAKO) were equally prepared and applied. On the next day, endogenous peroxidase was blocked for 15 min in a solution of 3% H_2_O_2_ (#31642, Sigma-Aldrich) prepared in H_2_O. An avidin/biotin-based peroxidase system (#VEC-PK-6100, Vector Laboratories, Newark, CA, USA) was applied to all slides in accordance with the manufacturer’s instructions and incubated in a wet chamber at RT for 1 h. Afterwards, secondary antibodies (Appendix A) were diluted at a ratio of 1:300 in PBS and incubated on the tissue samples for 1 h at RT. Nuclear ERα immunohistochemistry was performed with the Vector^®^ NovaRED^TM^ substrate Kit (#VEC-SK-4800, Vector Laboratories) and counterstained with methyl green (#H-3402, Vector Laboratories) as previously described for nuclear localization of ERβ [81]. LCN2 expression sites were detected with the 3,3′-diaminobenzidine tetrahydrochloride (DAB) chromogen (#D4293-50SET, Sigma-Aldrich) and counterstained with hematoxylin (#CS70030-2, Dako). Finally, tissue samples were dehydrated (increasingly graded ethanol, xylene) and mounted with DPX (#06522, Sigma-Aldrich).

#### 4.5.3. Immunofluorescence Staining

If formalin-fixed, paraffin-embedded slides were used for immunofluorescent staining, they were treated equally as for immunohistochemistry until incubation with different primary antibodies (LCN2: #AF3508, R&D System, 1:40; STAR: #12225-1-AP, ProteinTech Germany GmbH, Planegg-Martinsried, Germany, 1:50) or negative controls (normal goat IgG control, AB-108-C, R&D Systems; rabbit IgG control, #30000-0-AP, ProteinTech) overnight at 4 °C. 5% normal donkey serum (#ab7475, Abcam, Cambridge, UK) was used in the blocking solution (compare Section 4.5.2). Samples were protected from light during the consecutive staining steps to prevent unnecessary loss of fluorescent signals. Fluorescence-conjugated secondary antibodies (Appendix A), diluted at a ratio of 1:300 in PBS, were applied on tissue samples for 1 h at RT. Subsequently, counterstaining was performed with 4,6-diamidino-2-phenylindole dihydrochloride (DAPI) (#D1306, Thermo Fisher Scientific). Slices were mounted in aqueous PermaFluor^TM^ mounting medium (#TA-030-FM, Thermo Fisher Scientific). Control slides were treated equally except for incubation with H_2_O instead of a primary antibody. Stained slides, which gradually lose intensity after a couple of days, were stored in the dark at 4 °C for a short period of time prior to stable documentation (see Section 4.5.4).

#### 4.5.4. Imaging

Images were acquired with a Nikon Eclipse E80i fluorescence microscope equipped with the NIS-element Vis software (Version 3.22.01, Nikon, Tokio, Japan). Selected slides were further scanned with a NanoZoomer (#C13220-04, Hamamatsu, Naka-ku, Japan) and viewed with NDP.view2 software (version U12388-01, Hamamatsu).

### 4.6. Data Analysis

All calculations were performed using Excel v16 (Microsoft Corporation, Redmond, WA, USA). Protein expression was quantified densitometrically using ImageJ (software version 1.52n) and plotted relative to the protein loading control. Statistical analysis was performed with GraphPad Prism v.8.0 (GraphPad Software, Inc., La Jolla, CA, USA). Gaussian distribution was determined with Shapiro–Wilk tests. If normality could be assumed, a Student’s *t*-test was employed, otherwise, a non-parametric Mann–Whitney Test was chosen. For data that could be influenced by two factors (sex and genotype), two-way analysis of variance (ANOVA) was performed (i.e., Figure 1C and Figure 3D). Data are presented as means ± standard deviation (SD). Statistical significance between groups was assumed if probability values were below 0.05 (*p* < 0.05). Different significance levels are indicated by asterisks: * *p* < 0.05, ** *p* < 0.01, *** *p* < 0.001.

## 5. Conclusions

The aim of this study was to investigate whether the absence of ERα alters the expression of LCN2 in various non-reproductive and reproductive organs using an *Esr1*-deficient mouse strain. Possible tissue-specific changes in LCN2 were examined in both male and female animals to highlight whether there is a sex-specific difference in LCN2 expression. The data obtained in our study indicate that the expression of LCN2 in some non-reproductive and reproductive tissues is *Esr1*-dependent. While knockout of *Esr1* resulted in significantly increased expression of LCN2 in the testes and ovaries, the female lungs, as a non-reproductive organ, showed significantly lower expression of LCN2. Moreover, in some non-reproductive organs, there were no changes in LCN2 expression due to *Esr1* knockout, nor between the two sexes. Since ERα is not physiologically expressed in the lungs and kidney, it is most likely that the regulation of LCN2 is not primarily driven by ERα in these organs, but rather by systemic parameters. These differences suggest that the regulation of LCN2 is tightly controlled in each tissue. Further studies are now necessary to unravel the precise molecular pathways by which ERα impacts LCN2 expression in different tissues.

## Figures and Tables

**Figure 1 ijms-24-09280-f001:**
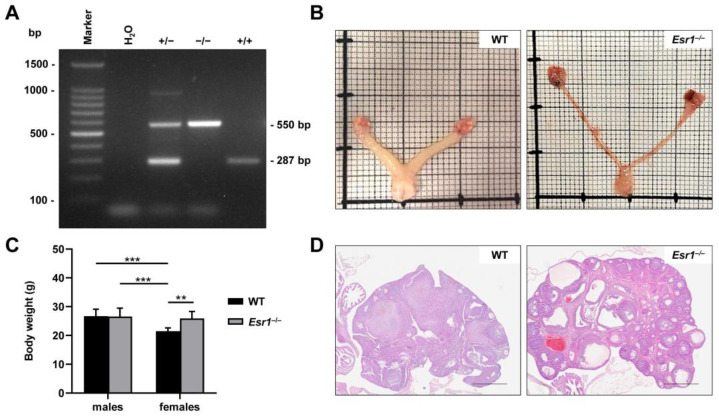
Depletion of estrogen receptor 1 (*Esr1*) in mice. (**A**) Traditional genotyping PCR was performed in wild-type (WT, +/+), *Esr1* null (−/−), and heterozygous *Esr1* mice (+/−). Top bands (550 bp) represent the depleted and bottom bands (280 bp) of the WT alleles. Heterozygous animals displayed both alleles. (**B**) Representative reproductive tracts from adult WT and knockout females were placed on scale paper to examine phenotypical changes. (**C**) Total body weights of WT (female n = 9, male n = 8) and *Esr*1^−/−^ mice (female n = 6, male = 8) were measured and are shown as means ± SD. For statistical analysis, a two-way ANOVA was performed. Significant differences between groups are indicated by asterisks: ** *p* < 0.01, *** *p* < 0.001. (**D**) Hematoxylin and eosin staining was performed for histological analysis of female ovaries taken from WT and *Esr1* null mice. Scale bars: 500 µm.

**Figure 2 ijms-24-09280-f002:**
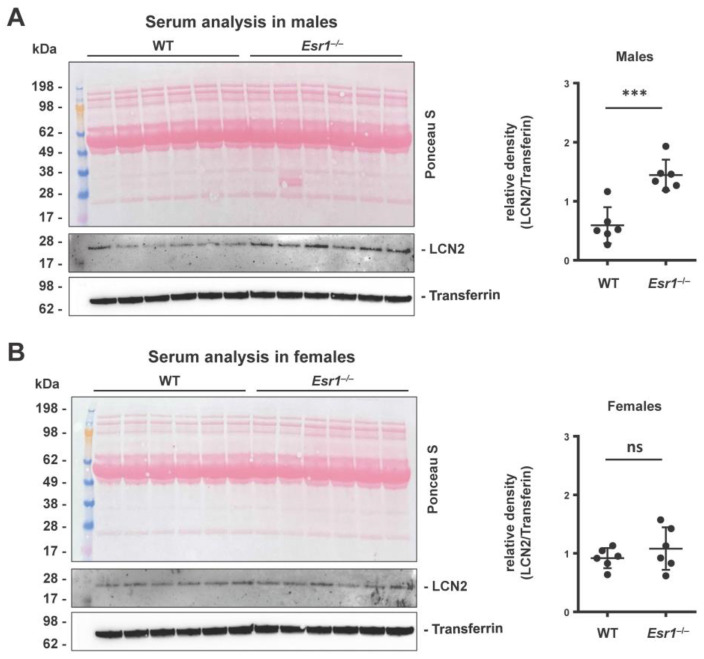
Analysis of serum LCN2 quantities in wild-type (WT) and *Esr1* null mice. Serum LCN2 levels were determined in (**A**) male and (**B**) female mice by Western blot analysis. Ponceau S stain and probing with transferrin were used to demonstrate equal loading. Protein expression of LCN2 was quantified densitometrically and is shown relative to transferrin expression. The Student’s *t*-test was used for statistical analysis. Significant differences between groups are indicated with asterisks: *** *p* < 0.001. Data are expressed as mean ± SD and non-significant results are marked with ns.

**Figure 3 ijms-24-09280-f003:**
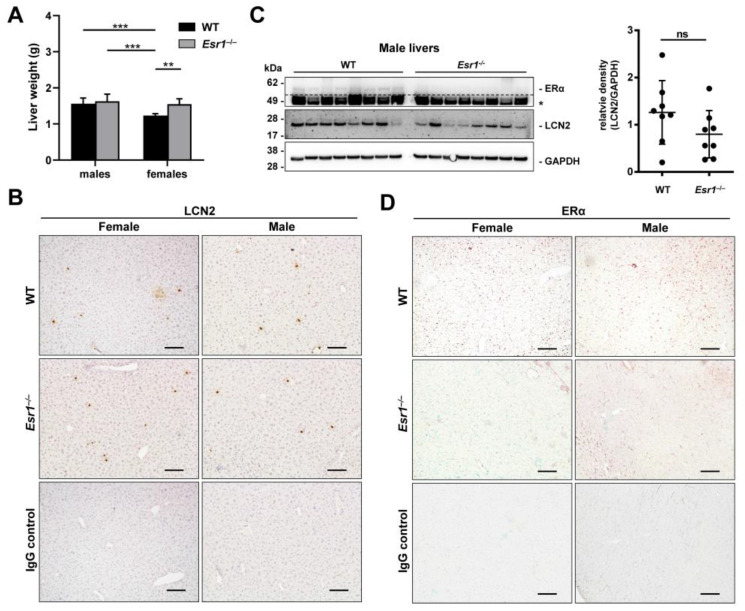
ERα and LCN2 expression in liver tissue. (**A**) Liver weights of wild-type (WT, male: n = 8, female: n = 9) and *Esr1* null mice (male: n = 8, female: n = 6) mice. Statistical analysis was conducted by performing a two-way ANOVA. Significant differences between groups are marked with asterisks: ** *p* < 0.01, *** *p* < 0.001. Data are displayed as means ± SD. (**B**) Paraffin-embedded WT (female n = 9, male n = 8) and *Esr1* knockout mice (female n = 6, male n = 8) livers were subjected to staining and protein analysis. Immunohistochemical detection of LCN2 was performed in liver tissues using 3,3′-diaminobenzidine (DAB) as a chromogen. The sections were counterstained with hematoxylin. (**C**) Male liver protein extracts were subjected to Western blot analysis for detection of ERα and LCN2. GAPDH was used to ensure equal protein loading. Unspecific bands are marked with asterisks (*) and are separated from specific bands by dashed lines. Protein expression of LCN2 was quantified densitometrically and plotted relative to GAPDH expression. The Student’s *t*-test was used for statistical analysis. Data are displayed as means ± SD and non-significant results are marked with ns. (**D**) Nuclear expression of ERα was analyzed in liver tissues using a NovaRed-HRP substrate. The sections were counterstained with methyl green. All magnifications in (**B**,**D**): 100×; scale bars: 100 µm.

**Figure 4 ijms-24-09280-f004:**
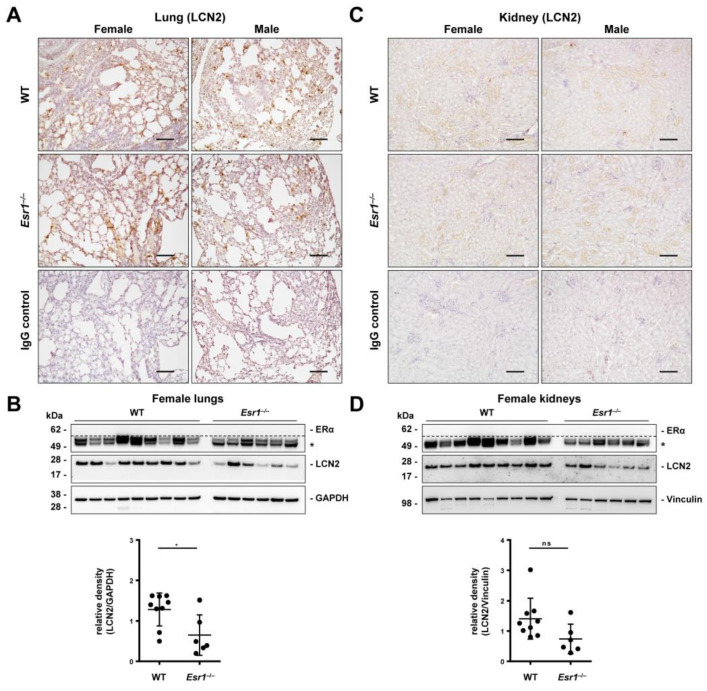
LCN2 and ERα expression in the lung and kidney. Tissues from wild-type (WT, female: n = 9, male: n = 8) and *Esr1* knockout (female: n = 6, male: n = 8) mice were either embedded in paraffin for immunostaining or processed for protein analysis. Immunohistochemistry staining was performed to localize LCN2 expression in (**A**) lung and (**C**) kidney tissue. Isotope-specific negative control IgG staining was performed to document antibody specificity. Magnifications: 100×; scale bar: 100 µm. LCN2 and ERα protein expression was analyzed in (**B**) lung and (**D**) kidney tissues by Western blot analysis in WT and *Esr1* null mice. The expression of GAPDH (lungs) and vinculin (kidneys) was used to demonstrate equal protein loading. Unspecific bands are marked with asterisks (*) and are separated from specific bands by dashed lines. Protein expression of LCN2 was quantified densitometrically and plotted relative to the expression of the loading control. Mann–Whitney test (lungs) or Student’s *t*-tests (kidney) was used for statistical analysis. Significant differences between groups are indicated with asterisks: * *p* < 0.05. Data are displayed as means ± SD and non-significant results are marked with ns.

**Figure 5 ijms-24-09280-f005:**
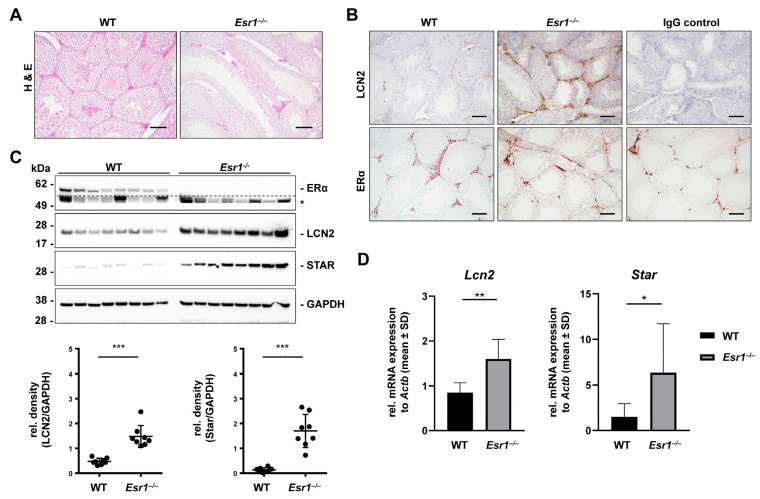
*Esr1*-deficient testes show increased LCN2 and STAR expression. Testes from wild-type (WT, n = 8) and *Esr1*^−/−^ null mice (n = 8) were either embedded in paraffin for immunostaining or processed for protein and RNA analyses. (**A**) Hematoxylin and eosin (H & E) staining was performed to acquire a histological overview. (**B**) Testes sections were subjected to LCN2 or ERα immunohistochemistry and counterstained with either hematoxylin (LCN2) or methyl green (ERα). Magnifications: 100×; scale bars: 100 µm. (**C**) LCN2, ERα, and STAR protein expression was analyzed by Western blot analysis. Probing with a GAPDH-specific antibody was used to demonstrate equal protein loading. Unspecific bands are marked with asterisks (*) and are separated from specific bands by dashed lines. Protein expression of LCN2 and STAR was quantified densitometrically and plotted relative to GAPDH expression. Mann–Whitney tests (LCN2) or Student’s *t*-tests (STAR) were used for statistical analysis. Significant differences between groups are indicated with asterisks: *** *p* < 0.001. Data are displayed as means ± SD. (**D**) Results were validated by RT-qPCR. Data are displayed as mean ± SD. For statistical analysis, a Student’s *t*-test was conducted. Significant differences between groups are marked with asterisks: * *p* < 0.05, ** *p* < 0.01.

**Figure 6 ijms-24-09280-f006:**
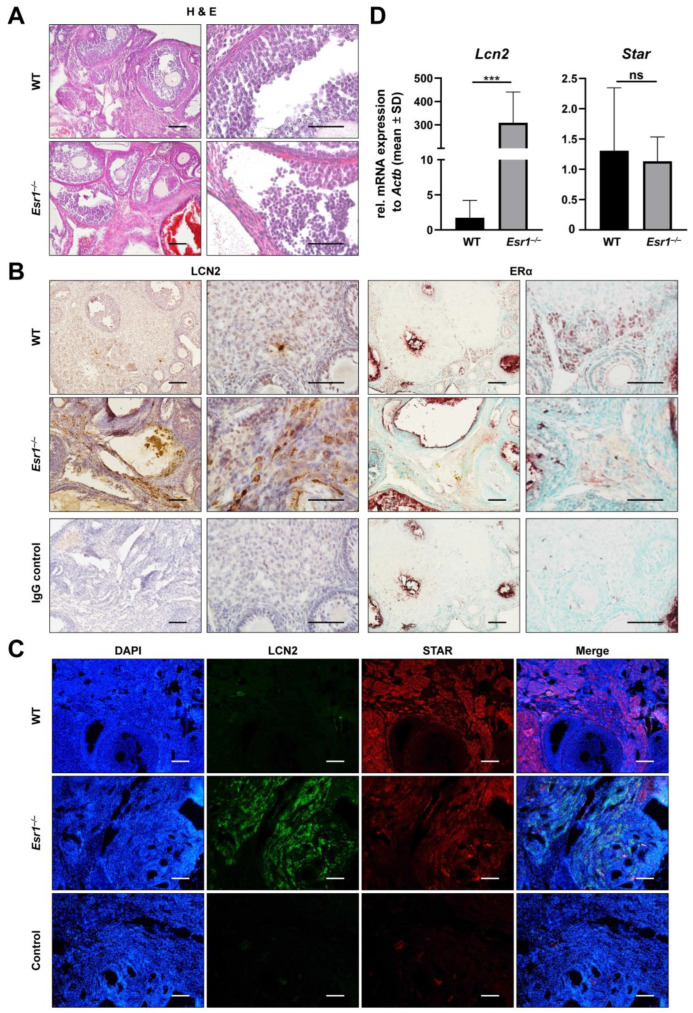
Increased LCN2 expression in *Esr1*-deficient ovaries. Ovaries from wild-type (WT, n = 8) and *Esr1* null mice (n = 6) were dissected and either embedded in paraffin or prepared for RNA analysis. (**A**) Hematoxylin and eosin (H & E) staining showed characteristic histological changes in *Esr1*-deficient compared to WT ovaries. (**B**) Immunohistochemical staining for LCN2 and ERα was performed. Sections were counterstained with hematoxylin (LCN2) or methyl green (ERα). Magnifications: 100×, 400×; scale bars: 100 µm, 50 µm. (**C**) Tissue samples were subjected to immunofluorescence double staining for LCN2 and STAR. Magnifications: 100×; scale bars: 100 µm. (**D**) *Lcn2* and *Star* expression was analyzed in ovarian tissues using RT-qPCR. Data are displayed as means ± SD. In case of normal distribution, Student’s *t*-tests were chosen for statistical analysis, otherwise a Mann–Whitney test was performed. Significant differences between groups are marked with asterisks: *** *p* < 0.001.

## Data Availability

Representative data of individual experiments are shown in this publication. However, additional data and replicates of depicted experiments are available on reasonable request from the corresponding authors.

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
