# Peer review of "Expression Analysis of Lipocalin 2 (LCN2) in Reproductive and Non-Reproductive Tissues of Esr1-Deficient Mice"

_ijms, 2023, doi:10.3390/ijms24119280_

Round 1

Reviewer 1 Report

1.     The manuscript needs some English language corrections to maintain the flow of text for example in the Abstract’s second line and so on.

2.     In line number 219-221, the paragraph that discusses Esr1-deficient male animals, it is unclear whether macroscopic changes are related to male or female infertility. The sentence can be rephrased for better understanding.

3.      Line number 438 and 439 needs rephrasing for clarity.

4.     The authors themselves mention that they have used total knockout mouse strain compared with organ-specific knockouts are systemic interactions that are not separately analyzed in this study, hence they were unable to find direct comparisons between different organ systems.

5.     The conclusion part should be specific and assertive on its own conception not vague, otherwise what is the purpose of the study if it does not lead towards a direction? Throughout the manuscript, a sex (reproductive) and tissue-specific (non-reproductive) correlation of LCN2 (Lipocalin 2) has been maintained with ERα (estrogen receptor-α) expression levels individually in Knockouts and Wild types, but the results are not in accord. This should be clearly addressed in the last. 

Moderate editing of English language

Author Response

Dear Reviewer 1,

many thanks for your time you spent in reading our manuscript. Please find attached our comments to your suggestions in the attached pdf-file.

Regards

Ralf Weiskirchen

Reviewer 2 Report

This is an interesting paper analysing the expression of lipocalin 2 in selected tissues of Esr1-deficient mice. In my opinion, however, it has at least two weak points.

1.       The results obtained by Western blot method were not statistically analysed. Despite this, the authors describing the results obtained with this method write about an increase or decrease in protein expression (e.g. lines 150-152, 184, 205-206, 256-257, 259 etc.). I think this may be a misinterpretation.

2.       The study of protein expression in the liver, lungs, kidneys and spleen is based on a small number of biological replicates - 2 or 3.

Minor remarks

1.       Please note the subsection numbering in section 4.5 Histology.

2.       Please complete the information about the reagent manufacturers (e.g. line 647).

3.       Line 673: Figure C should be mentioned instead of Figure 1B.

Author Response

Dear Reviewer 2,

many thanks for your time you spent in reading our manuscript. Please find attached our comments to your suggestions in the attached pdf-file.

Regards

Ralf Weiskirchen

Round 2

Reviewer 2 Report

I accept the authors' defense and appreciate corrections in the MS. Please ensure that figures are properly cited (e.g. line 437 and 465).